# The Utilization of Alkali-Activated Lead–Zinc Smelting Slag for Chromite Ore Processing Residue Solidification/Stabilization

**DOI:** 10.3390/ijerph18199960

**Published:** 2021-09-22

**Authors:** Lin Yu, Lu Fang, Pengpeng Zhang, Shujie Zhao, Binquan Jiao, Dongwei Li

**Affiliations:** 1State Key Laboratory for Coal Mine Disaster Dynamics and Control, Chongqing University, Chongqing 400044, China; YL518@cqu.edu.cn (L.Y.); fl2018cqu@163.com (L.F.); 20172002011t@cqu.edu.cn (P.Z.); 20162002019t@cqu.edu.cn (S.Z.); 2School of Resources and Safety Engineering, Chongqing University, Chongqing 400044, China

**Keywords:** solidification/stabilization, cotreatment, lead–zinc smelting slag, alkali-activated materials, chromite ore processing residue

## Abstract

Lead–zinc smelting slag (LZSS) is regarded as a hazardous waste containing heavy metals that poses a significant threat to the environment. LZSS is rich in aluminosilicate, which has the potential to prepare alkali-activated materials and solidify hazardous waste, realizing hazardous waste cotreatment. In this study, the experiment included two parts; i.e., the preparation of alkali-activated LZSS (pure smelting slag) and chromite ore processing residue (COPR) solidification/stabilization. Single-factor and orthogonal experiments were carried out that aimed to explore the effects of various parameters (alkali solid content, water glass modulus, liquid–solid ratio, and initial curing temperature) for alkali-activated LZSS. Additionally, compressive strength and leaching toxicity were the indexes used to evaluate the performance of the solidified bodies containing COPR. As a result, the highest compressive strength of alkali-activated LZSS reached 84.49 MPa, and when 40% COPR was added, the strength decreased to 1.42 MPa. However, the leaching concentrations of Zn and Cr from all the solidified bodies were far below the critical limits (US EPA Method 1311 and China GB5085.3-2007). Heavy-metal ions in LZSS and COPR were immobilized successfully by chemical and physical means, which was detected by analyses including environmental scanning electron microscopy with energy-dispersive spectrometry, Fourier transform infrared spectrometry, and X-ray diffraction.

## 1. Introduction

Lead–zinc smelting slag (LZSS) is a nonferrous metal smelting residue that belongs to China’s national hazardous waste list. It is considered as a hazardous waste containing heavy metals. Due to the fast development of the nonferrous metal industry, much higher levels of smelting residues were produced, and most smelting residues previously were dumped without safe disposal [1,2,3]. The long-term storage of massive smelting waste causes great damage to the ecosystem and threatens human health [4,5]. Therefore, the treatment and resource utilization of smelting slag have aroused attention.

At present, various technologies have been used to dispose LZSS, including solidification/stabilization [4,6], chemical reduction [2], metal recovery [5], and the resource-utilization method [7]. Solidification/stabilization is a relatively mature technology for harmless pretreatment and waste reuse, and is widely used in waste-material disposal [8,9,10]. It converts pollutants into a less-soluble form and immobilizes pollutants by creating a persistent matrix to encapsulate them [11]. The technology immobilizes contaminants by geopolymerization, the reaction mechanism of which mainly includes dissolution, migration, gelation, reorganization, polymerization, and hardening of Al and Si precursor species [12]. Materials that were rich in aluminosilicate; i.e. blast furnace slag, rice husk ash, fly ash, and natural pozzolans were often used as solidified precursors [13]. Moreover, some researchers have proven that solidified precursors could be replaced by partial smelting slag [14,15]. Albitar [16] replaced fly ash with granulated lead smelter slag. The results indicated that the compressive strength of fly ash geopolymer concrete decreased as the proportion of smelter slag to replace fly ash binder increased. However, the compressive strength was always higher than 30 MPa. Similarly, Nath [7] added zinc slag into fly ash to improve strength. The results showed that toxic-metal leaching was within the permissible limit, and the strength of samples was up to 71 MPa with 100% addition of zinc slag. Furthermore, Zhang et al. [17] prepared alkali-activated materials with LZSS. The compressive strength of solidified products reached 96.14 MPa with a removal rate of heavy metals above 90%, indicating that LZSS had a good self-cementing effect. As described, alkali-activated LZSS could solidify/stabilize hazardous waste due to the excellent performance in compressive strength and heavy-metal-leaching toxicity.

As important industrial raw materials, chromium and chromium salts are mainly used in chemical metallurgy, tanning, electroplating, and other industries [18,19]. Chromite ore processing residue (COPR) is a byproduct generated in producing chromium and chromium salts by high-temperature (approximately 1200 °C) roasting. COPR does great harm to the environment and human health due to its high content of hexavalent chromium, which has high mobility, mutagenicity, carcinogenicity, and teratogenic effects [20,21]. Since high-calcium roasting generated up to three tons of residue per ton of sodium dichromate product, it has been banned in China since 2013 [22]. Nevertheless, other roasting methods; i.e. the less-calcium or noncalcium roasting technologies that produce less residue, still threaten the environment [22,23]. As proved by previous studies, solidification/stabilization is an excellent method to deal with COPR [24]. Huang et al. [25] solidified COPR by blast furnace slag-based geopolymer. The results showed that 60% COPR addition had effective solidification (compressive strength was 18.52 MPa, and hexavalent chromium leaching concentration was 1.553 mg/L). Sun et al. [26] immobilized COPR with a metakaolin-based geopolymer-added Na_2_S. The COPR content was 20%, the mechanical strength was above 42 MPa, and the leaching concentration of hexavalent chromium was less than 5 mg/L. Generally, industrial wastes or calcined clays are used to solidify COPR, but few studies have focused on hazardous wastes as curing agents. The utilization of alkali-activated LZSS to solidify/stabilize COPR provides a new way for the cotreatment of hazardous wastes, which can reduce the harm of COPR and reuse LZSS.

In this study, alkali-activated LZSS was used as the binder to solidify/stabilize COPR. The compressive strengths of alkali-activated LZSS using different alkali solid contents, water glass moduli, initial curing temperatures, and liquid–solid ratios were studied. The optimal combination was obtained by the orthogonal experiment. The heavy-metal extraction toxicity and compressive strength of the solidified bodies with different content of COPR (LZC) were investigated. Ultimately, the immobilization mechanism of heavy-metal ions was analyzed with an environmental scanning electron microscope with energy-dispersive spectrometry (ESEM-EDS), Fourier transform infrared spectrometry (FTIR), and X-ray diffraction (XRD).

## 2. Materials and Methods

### 2.1. Materials

The raw materials mainly included LZSS, COPR, and the alkali activator. The irregularly shaped LZSS was sampled from a lead–zinc smelter located in Yunnan Province, China. The COPR was purchased from a chemical factory in Chongqing, China. The alkali activator was prepared using water glass (Na_2_O 3.3SiO_2_), sodium hydroxide (NaOH), and deionized water. The chemical composition of the LZSS and COPR was analyzed using X-ray fluorescence spectra, and the results are shown in Table 1.

### 2.2. Experimental Methods

#### 2.2.1. Preparation of Alkali-Activated LZSS Samples

Initially, the untreated LZSS was dried for about 6 h in an oven at 105 °C. The dried LZSS was ball-milled to reduce the particle size, and the product was passed through a 200 mesh sieve. Next, the LZSS and the alkali activator were mixed and stirred for 5 min and then poured into the mold (20 mm × 20 mm × 20 mm). The alkali activator was prepared by mixing a certain proportion of deionized water, water glass, and sodium hydroxide, and was placed for 18 h. The samples were vibrated on a vibrating table to remove bubbles. Finally, the prepared samples were cured at the initial curing temperature for 1 day, and then the compressive strength was tested after curing at room temperature up to 7 days and 28 days [6]. Single-factor and orthogonal experiments were evaluated, taking the compressive strength of samples as an index. The single-factor experiment included alkali solid content, water glass modulus, initial curing temperature, and liquid–solid ratio; the arrangement is shown in Table 2. Furthermore, alkali solid content (7%, 8%, 9%), water glass modulus (1.3, 1.4, 1.5), and liquid–solid ratio (0.18, 0.19, and 0.20) were selected for the orthogonal experiment based on the single-factor experiment. The experimental arrangement is illustrated in Table 3. The samples were cured at the initial curing temperature of 35 °C (the optimal curing temperature in the single-factor experiment) for 24 h, and then the mold was cured at room temperature for 28 days.

#### 2.2.2. Stabilization/Solidification of COPR

Based on the different water absorption of raw materials, the sample with alkali solid content of 7%, water glass modulus of 1.5, and liquid–solid ratio is 0.2 was selected to solidify the COPR. The LZSS in the alkali-activated sample was replaced by different ratios of COPR. The proportions of COPR added were 0, 10%, 20%, 30%, 40% (the sample could not be stirred evenly, hence we set the liquid–solid ratio to 0.24). The solidified bodies with different added ratios of COPR were named LZC0–40, respectively. The COPR was required to be dried at 105 °C for 6 h before solidification.

#### 2.2.3. Compressive Strength Tests

Every three parallel samples were taken as a group, and the compressive strength of samples was measured with an electronic precision material testing machine (AG-250 kN IS, SHIMADZU, Kyoto, Japan). The results were averaged (relative error < 10%).

#### 2.2.4. Determination of Leaching Toxicity

Different leaching means were used to determine the extraction toxicity of the heavy metals in the raw materials and solidified samples. All leaching tests were performed immediately after 28 days of curing. The leaching methods included the sulfuric acid and nitric acid method (China HJ/T299-2007) and the toxicity characteristic leaching procedure (TCLP, US EPA Method 1311). In TCLP, the solid was crushed to pass through a 9.5 mm standard sieve. The extractant was prepared by diluting 5.7 mL glacial acetic acid to 1 L (diluted with distilled water), and the extractant kept the pH at 2.88 ± 0.05. The crushed sample was mixed with the extractant at a liquid-to-solid ratio of 20:1, and then placed in an oscillating flip device to shake for 18 ± 2 h. The oscillating flip device operated at a rotation speed of 30 ± 2 r/min. After shaking, the solid–liquid mixture was suction-filtered to obtain the filtrate for testing. In the sulfuric acid and nitric acid method, the crushing, shaking, and suction filtration steps were the same as in TCLP. However, some differences existed in the extractant and liquid–solid ratio. The extractant was prepared by diluting the mixture of sulfuric acid and nitric acid (mass ratio 2:1) to 1 L, and the extractant kept the pH at 3.20 ± 0.05. The liquid–solid ratio was 10:1. The heavy-metal ions in the two filtrates were detected using an atomic absorption spectrophotometer. The pH of the filtrate was adjusted to less than 2 with dilute nitric acid before detection.

#### 2.2.5. Characterization Analysis

The raw materials and solidified bodies were analyzed and characterized through XRD, FTIR, and ESEM-EDS. The microstructure and chemical composition were determined by ESEM-EDS (Thermo Scientific Quattro S, Waltham, MA, USA) at an accelerating voltage of 20 kV. The compound structure was analyzed via FTIR (Thermo Nicolet iS50, Waltham, MA, USA) with a wavenumber range of 400–4000 cm^−1^. The phase composition was observed via XRD (PANalytical X’Pert Powder, Almelo, The Netherlands) with the following conditions: CuK_α_ radiation, 40 kV, 30 mA.

## 3. Results and Discussion

### 3.1. Compressive Strength Analysis

#### 3.1.1. Single-Factor Experiment

(1)Alkali solid content

Alkali solid content refers to the mass percentage of alkali solid (Na_2_SiO_3_ in water glass and NaOH) and LZSS. Alkali content influences the compressive strength of alkali-activated LZSS. Fang et al. [27] proved that an increase in alkali content had a positive effect on strength for alkali-activated materials (alkali content 2–6%). As shown in Figure 1a, this study appraised the influence on compressive strength with different alkali solid contents; i.e., 6%, 8%, 10%, and 12%. The compressive strength of the samples increased as curing time increased. After 28 d, with the increasing of alkali solid content, the compressive strength showed a trend of first increasing and then decreasing. The maximum strength of the samples was obtained with 8% alkali solid content. The compressive strength of samples after 28 d increased by 41.65% (A1), 6.22% (A2), 16.38% (A3), and 4.38% (A4) compared to after 7 d, respectively. More Na and Si contributing to aluminosilicate gel formation was produced as alkali solid content increased, and the aluminosilicate gel could improve the compressive strength by hardening to form the alkali-activated binder. A similar conclusion was drawn in the study of Singh et al. [28]. However, when alkali solid content exceeded a certain limit, excessive sodium silicate may have impeded water evaporation and polymerization product formation during the process of polycondensation, affecting the development of compressive strength. Barbosa et al. [29] and Huang et al. [30] also confirmed this point.

(2)Water glass modulus

The water glass modulus has some impact on compressive strength. As illustrated in Figure 1b, different water glass moduli (1.0, 1.2, 1.4, 1.6, and 1.8) were selected. The compressive strength after 28 d was higher than that after 7 d. As the modulus increased, the water glass modulus first had a positive effect on the strength, and then had a negative effect. When the modulus was 1.4, the compressive strength reached its maximum value (7 d, 61.45 MPa; 28 d, 65.28 MPa). In general, a higher water glass modulus provided more soluble silicate (the raw material of Si–Al structure) and contributed to higher compressive strength, as demonstrated by Dimas et al. [31]. However, too high a modulus means more sodium silicate in cementation materials. The more sodium silicate added to the cementation materials, the lower the alkalinity, and the excess sodium silicate would reduce geological polymerization reaction activity, causing a decrease in the compressive strength. Huang et al. [25] and Cho et al. [32] reached the same conclusion.

(3)Initial curing temperature

The initial curing temperature is a vital factor for properties of alkali-activated materials. Zhang et al. [17] proved that low-temperature (30–50 °C) curing was more suitable for alkali-activated LZSS. This study focused on the influence of temperature (25 °C, 35 °C, 45 °C, 55 °C) on the strength of alkali-activated LZSS. As illustrated in Figure 1c, the influence of initial curing temperature on compressive strength showed an insignificant trend after 28 d. From 25 °C (62.31 MPa) to 35 °C (65.28 MPa), the strength increased by 4.77%, and from 35 °C (65.28 MPa) to 45 °C (63.86 MPa), the strength decreased by 2.18%. A moderate temperature rise is conducive to the dissolution of aluminosilicates, while too high a curing temperature will result in the reduction of strength due to the reduction of geopolymer basic media [33]. In addition, a higher temperature may result in the formation of pores due to the rapid evaporation of water, affecting strength [34].

(4)Liquid–solid ratio

The liquid–solid ratio refers to the ratio of water (deionized water and water in water glass) to total solids (including LZSS, Na_2_SiO_3_, and NaOH). Conspicuously, water content plays a significant role in the alkali-activated process [35]. A suitable amount of deionized water can be used as a medium, which is conducive to the hydrolytic condensation of the material [35]. As shown in Figure 1d, the compressive strength after 28 d was slightly higher than that after 7 d. When the liquid–solid ratio increased, the compressive strength after 7 d and 28 d both declined. The samples for which the liquid–solid ratio was less than 0.18 could not be formed due to uneven mixing. In contrast, an excessive liquid–solid ratio led to a decrease in compressive strength. On the one hand, excess water caused many pores when evaporated from the alkali-activated LZSS, reducing the strength. On the other hand, in the specimen-molding process, water was extruded from the corresponding excess liquid–solid ratio test pieces. This process also led to the loss of some alkaline substances, which affected aluminosilicate dissolution and had a negative impact on the strength [36,37].

#### 3.1.2. Orthogonal Experiment

Based on the results of single-factor experiments, a three-factor (liquid–solid ratio, water glass modulus, and alkali solid content) orthogonal experiment was carried out to prepare the LZSS cementitious materials. The specific preparation procedure was described in Section 2.2.1, and the experimental results are shown in Table 3.

From the R value (Table 3), the compressive strength was influenced in the following order: alkali solid content (A) > water glass modulus (B) > liquid-solid ratio (C). Given the above, the optimal combination could be A1B3C2, at which time the alkali solid content was 7%, the water glass modulus was 1.5, the liquid–solid ratio was 0.19, and the initial curing temperature was 35 ℃, according to the results for the optimal individual parameters. With the above optimal combination, the strength of the alkali-activated LZSS reached 84.49 MPa after 28 d.

### 3.2. Stabilization/Solidification of COPR

#### 3.2.1. Sample Strengths

A downtrend in strength for different solidified bodies can be seen in Figure 2. The COPR content increased from 10% to 40%, and the compressive strength decreased by 9.24% (10% COPR), 18.11% (20% COPR), 68.41% (30% COPR), and 98.30% (40% COPR), respectively. With 40% COPR addition, the compressive strength was 1.42 MPa, which met the landfill requirements (>0.35 MPa) [8]. Meanwhile, when the COPR content was 30%, the compressive strength was 26.37 MPa, achieving the strength requirements for construction purposes (>10 MPa) [38]. After 28 d, the compressive strengths for different additions of COPR increased by 8.32% (10% COPR), 27.44% (20% COPR), 84.28% (30% COPR), and 57.78% (40% COPR) compared with those after 7 d, respectively. Moreover, the strength growth rate increased first and then decreased with the addition of COPR. The sharp drop in strength, which can be seen in Figure 2, might have been due to the difference in alkali-activation activity and composition of the two materials [39].

#### 3.2.2. Leaching Toxicity Analysis

The leaching concentrations and critical limits of heavy metals in raw materials and solidified bodies are shown in Table 4. The leaching concentrations of Zn and Cr in LZSS were below the critical limits (EPA critical limits for the TCLP method, and GB5085.3-2007 critical limits for the sulfuric acid and nitric acid method). However, the leaching concentration of Zn in the TCLP method reached up to 124.63 mg/L. Moreover, the leaching concentration of Cr in COPR exceeded the criterion thresholds of the EPA (5 mg/L) [40,41] and GB5085.3-2007 (15 mg/L) [41]. The leaching concentrations and solidification efficiency of heavy metals were analyzed to evaluate the solidification performance. The solidification efficiency of Zn and Cr in solidified bodies (LZC) can be seen in Figure 3; it was calculated using the following formula:(1)$$\mathrm{Solidification}\;\mathrm{efficiency}\;\left(\%\right)=1\;-\;\frac{\mathrm{the}\;\mathrm{leaching}\;\mathrm{concentration}\;\mathrm{of}\;\mathrm{solidified}\;\mathrm{bodies}}{\mathrm{the}\;\mathrm{leaching}\;\mathrm{concentration}\;\mathrm{of}\;\mathrm{raw}\;\mathrm{materials}}$$

The leaching concentration of Zn in the sulfuric acid and nitric acid method was much lower than that in the TCLP method, which was compatible with the conclusion drawn by Mao et al. [4]. Meanwhile, the solidification efficiency of Zn in different leaching methods showed a difference. The Zn leaching concentration of LZC10 was lower than that of LZC0 in both leaching methods. This may have been due to the higher content of Al_2_O_3_ in COPR (29.88%) optimizing the ratio of LZSS; however, excessive content leads to incomplete dissolution of aluminosilicate and hinders the condensation reaction, thus causing the leaching concentration to gradually increase. After solidification, the leaching concentrations of Cr in both leaching methods were far below the critical limits. Meanwhile, with the increase in COPR content, the Cr leaching concentration increased. This was due to the increase in chromium content in the LZC with the increase in COPR content, while the compressive strength of the LZC was decreasing, which increased the leaching concentration of Cr [39]. Moreover, the solidification efficiency of Cr in both methods was over 95%, indicating an excellent solidification effect.

### 3.3. XRD

The phase compositions of LZSS and COPR were analyzed through the XRD images shown in Figure 4. LZSS is water-quenched residue slag that is produced by smelting lead–zinc metal at a high temperature. It is a mostly glassy amorphous substance with a less-crystalline state. The existence of two amorphous humps between 10°–40° could be seen in LZSS. In addition, LZSS mainly contained three kinds of crystals; i.e. zinc aluminum iron oxide (PDF# 82-1048), zinc oxide (PDF# 77-0191), and wustite (PDF# 74-1880). In comparison, COPR had relatively legible crystal peaks, mainly involving magnesiochromite (PDF# 87-1175), magnesium chromium oxide (PDF# 77-0007), and magnesium aluminum iron oxide (PDF# 71-1235).

The samples with different contents of COPR were characterized by XRD. Under the action of alkali excitation, the Si, Ca, and Al rich in LZSS underwent dissolution and reorganization to form an alkali-activated cementitious material (LZC0). In Figure 5, the diffraction peaks of zinc iron oxide (PDF# 73-1963) and zinc oxide (PDF# 77-0191) can be observed for LZC0. With the addition of COPR of higher crystallinity, the magnesiochromite, ferroan (PDF# 09-0353), and magnesium aluminum iron oxide (PDF# 71-1235) from the COPR were still retained in the LZC10-40. As the COPR content increased, the peak strength of the main crystalline phase magnesium chromite gradually increased, which may have led to incomplete dissolution of aluminosilicate, thus hindering the condensation reaction and leading to a decrease in compressive strength [24].

### 3.4. FTIR

The FTIR images of samples (LZC0-40) are presented in Figure 6. The broad absorption peaks at approximately 3400–3500 cm^−1^ were associated with an -OH asymmetric stretching vibration [42]. Meanwhile, the absorption peaks in the range of 1600–1650 cm^−1^ were due to the bending vibration of H-O-H [43]. These suggested the existence of free and bound water. The characteristic absorption peaks of about 1410–1480 cm^−1^ represented carbonates [44]. Their formation was associated with the participation of carbon dioxide in the process of alkali activation [45]. Moreover, the peaks at nearly 1000 cm^−1^ indicated an asymmetric stretching vibration of the Si-O-T (T = Si, Al) bond [46]. The Si-O-T peak shifted towards the low wavenumber direction with the addition of COPR. This might have been due to the Si/Al ratio of samples reduced with the admixture of COPR (due to its high content of Al_2_O_3_). This might mean some Si-O were substituted by Al-O, forming more [AlO_4_]^−^ tetrahedrons [47]. Moreover, the [AlO_4_]^−^ tetrahedron was negatively charged, and some Na^+^ and Ca^2+^ might have been involved in charge balance to balance the charge of the system [17]. In addition, studies have shown that Si-O-T may be transformed into nonbridging oxygen (T-O-Na^+^) during the alkali-activation process, thereby providing sites for the substitution of heavy-metal ions [48]. Meanwhile, Na^+^ and Ca^2+^ (charge-balance ions) might have been replaced by heavy-metal ions (Zn and Cr), which would indicate chemical bonding in the solidification process [6,44]. The peaks at band 400–549 cm^−1^ represented Si-O and Al-O bending vibrations [49]. The peak in the vicinity of 444.12 cm^−1^ shifted towards the direction of high wavenumbers, indicating that the heavy-metal ions might have been partially fixed in the structure through chemical bonding [25].

### 3.5. ESEM-EDS

The microstructures of the raw materials and solidified bodies were analyzed by ESEM-EDS. It can be observed in Figure 7a that the LZSS was an irregular residue with edges and corners. This might benefit the formation of a network structure [4,50]. The size of the COPR particles varied, and there were a few lamellar structures and acicular structures (Figure 7b). As can be seen in the micrograph in Figure 7c, the solidified body without COPR (LZC0) had a dense structure, which was consistent with its high mechanical strength. Meanwhile, some white gelatinous material was attached to the surface of the dense structure. This might have been (C, N)-A-S-H gel, based on the analysis of the elemental composition of spot 1 in Table 5. The mapping image of area e (Figure 7g) showed mainly N-A-S-H gels [17]. The higher heavy-metal atomic content at spot 2 in Table 5 indicated that the heavy metals might have been fixed into the structure of geopolymer. Compared with LZC0, LZC30 had a looser structure, and more pores were formed on its surface, which led to a decrease in mechanical strength. The irregular structure in area f was similar to that of LZSS. This might be because the addition of COPR hindered the geological polymerization reaction. It also suggested the possibility of physical immobilization for heavy-metal ions. The content of Fe atoms in area f was much higher than that in area e. The presence of Fe in the alkali-activation reaction hindered the release of Si and Al in an alkaline environment, thereby hindering the process of alkali activation, resulting in a decrease in strength [51].

## 4. Conclusions

In this paper, alkali-activated LZSS was used as a binder to solidify and stabilize COPR. With compressive strength as an index, single-factor and orthogonal experiments were carried out, and suitable parameters were selected to solidify/stabilize COPR. We evaluated the performance of solidified bodies with COPR by compressive strength and heavy-metal leaching. The immobilization mechanism of heavy metals was analyzed by XRD, FTIR, and ESEM-EDS. The following conclusions were drawn.

The optimal experimental conditions were obtained via the single-factor and orthogonal experiments: for a liquid–solid ratio of 0.19, an alkali solid content of 7%, a water glass modulus of 1.5, and an initial curing temperature was 35 °C, the highest compressive strength of 84.49 MPa was reached.

The compressive strength and leaching of heavy metals of solidified samples are two significant indicators that are used to evaluate the potential use of solidification/stabilization techniques for different purposes. The compressive strength (28 d) of the LZC showed a downward trend with an increase in the COPR content (10–40%). When the content was 30%, the compressive strength was 28.35 MPa, which met the compressive-strength requirements for building materials (10 MPa). The leaching concentration of Cr in the LZC gradually increased with an increase in the COPR content, but all of them were under the safety limit of 5 mg/L; moreover, the solidification efficiency of Cr was more than 95%. These results showed that the LZC could be utilized as effective management of COPR waste, and could also be used for construction purposes.

The ESEM-EDS results demonstrated that the alkali-activated LZSS formed a dense structure, and the heavy-metal ions were physically immobilized in the geopolymer gel. However, the addition of COPR caused the destruction of the dense structure. The deviation of absorption peaks in the FTIR showed that metal cations (Na^+^, Ca^2+^) might have been replaced by heavy metals. In conclusion, alkali-activated LZSS could be used to solidify COPR, and heavy metals (Zn and Cr) were effectively immobilized by chemical and physical means.

## Figures and Tables

**Figure 1 ijerph-18-09960-f001:**
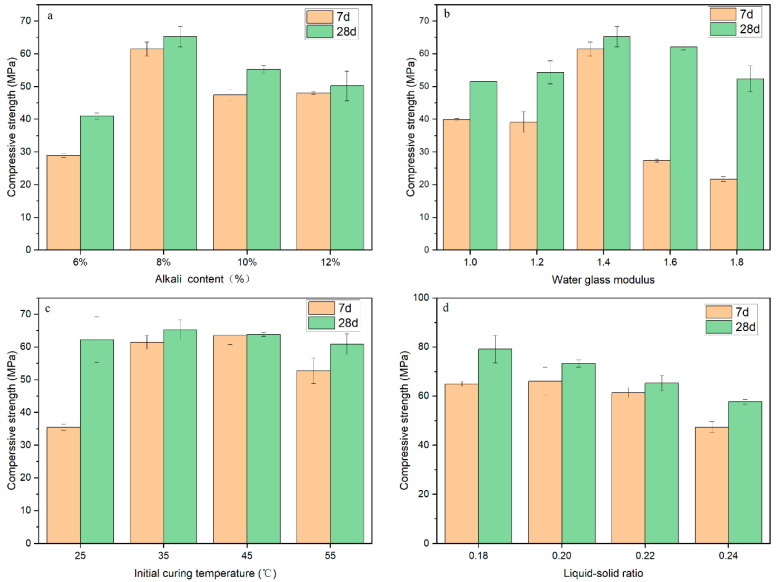
The compressive strength of different samples in the single-factor experiment: (**a**) alkali solid content; (**b**) water glass modulus; (**c**) initial curing temperature; (**d**) liquid–solid ratio.

**Figure 2 ijerph-18-09960-f002:**
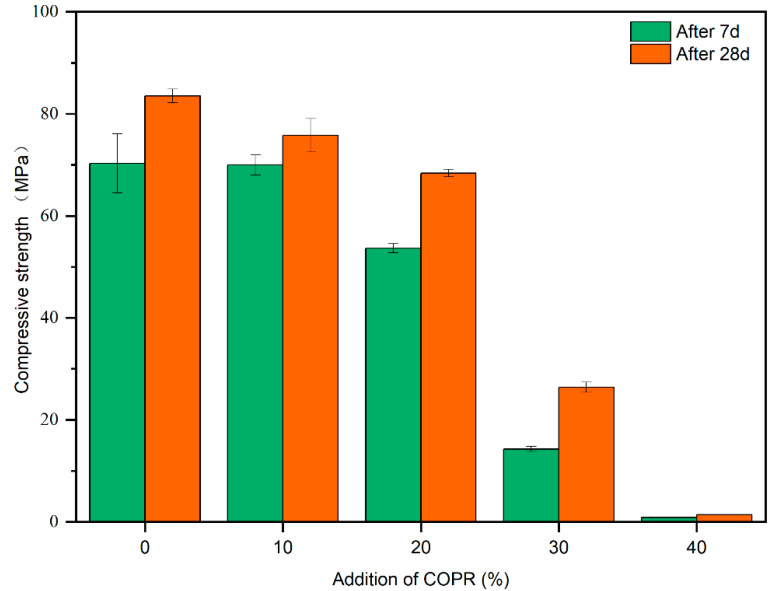
The compressive strengths of solidified bodies (LZC) with different admixtures of COPR.

**Figure 3 ijerph-18-09960-f003:**
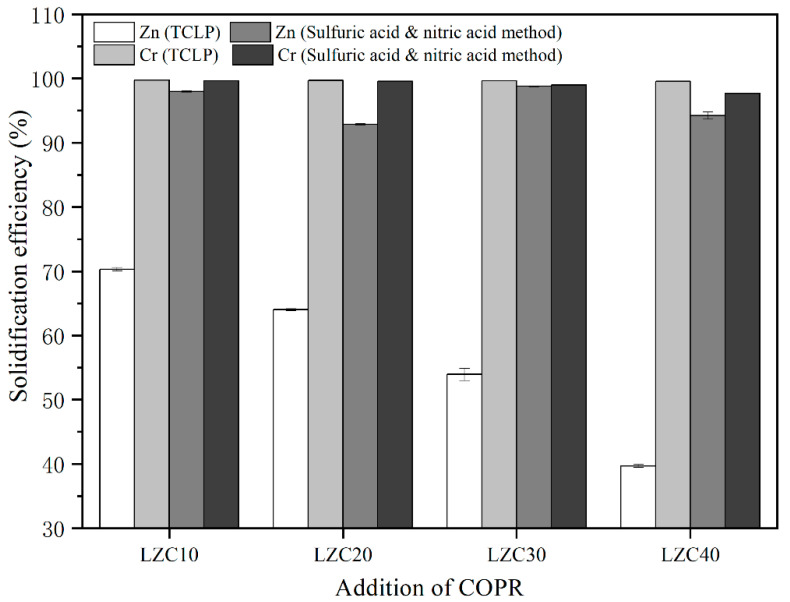
The solidification efficiency of heavy metals for LZC10–40 (10, 20, 30, 40, corresponding to the concentration of COPR).

**Figure 4 ijerph-18-09960-f004:**
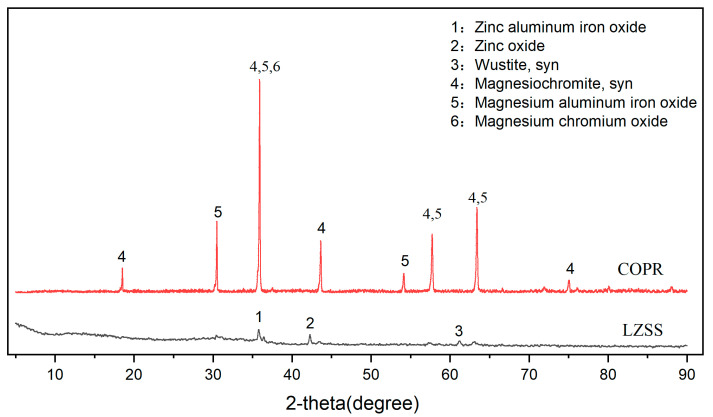
The XRD images of the raw materials (LZSS and COPR).

**Figure 5 ijerph-18-09960-f005:**
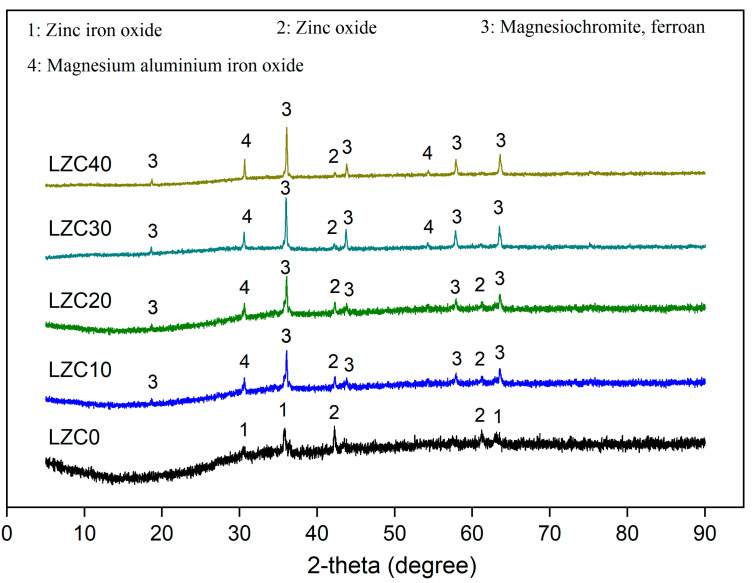
The XRD patterns of solidified bodies (LZC0–40).

**Figure 6 ijerph-18-09960-f006:**
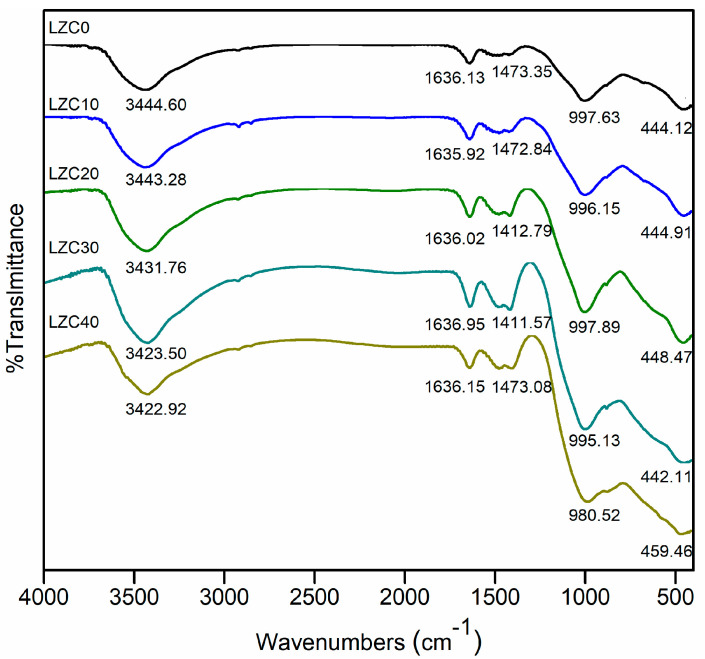
FTIR images of solidified bodies (LZC0–40).

**Figure 7 ijerph-18-09960-f007:**
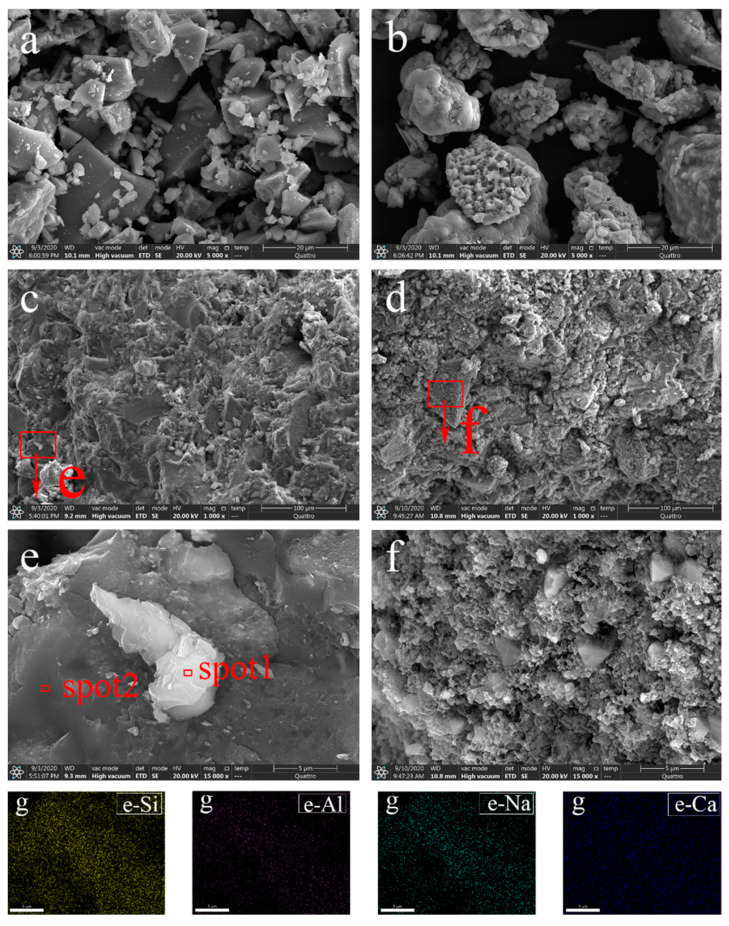
The ESEM micrographs of (**a**) LZSS, (**b**) COPR, (**c**) LZC0, (**d**) LZC30, (**e**) area e, and (**f**) area f; and (**g**) the mapping images for Si, Al, Na, and Ca in area e.

**Table 1 ijerph-18-09960-t001:** Chemical compositions of COPR and LZSS (wt %).

Materials	SiO_2_	Fe_2_O_3_	CaO	Al_2_O_3_	ZnO	MgO	SO_3_	Na_2_O	Cr_2_O_3_	TiO_2_	MnO	Others
LZSS	32.78	35.69	10.03	8.17	2.19	2.58	2.24	0.76	0.12	0.74	3.12	1.58
COPR	2.99	37.82	0.13	29.88	0.13	11.97	0.11	4.13	10.83	1.33	-	0.68

**Table 2 ijerph-18-09960-t002:** Single-factor experimental design.

ID	Alkali Solid Content (%)	Water Glass Modulus	Initial Curing Temperature (°C)	Liquid–Solid Ratio
A1	6	1.4	35	0.22
A2	8			
A3	10			
A4	12			
M1	8	1.0	35	0.22
M2		1.2		
M3		1.4		
M4		1.6		
M5		1.8		
T1	8	1.4	25	0.22
T2			35	
T3			45	
T4			55	
L1	8	1.4	35	0.18
L2				0.20
L3				0.22
L4				0.24

Note: alkali solid content: the mass ratio of alkali solid (Na_2_SiO_3_ and NaOH) and LZSS; liquid–solid ratio: the ratio of water (deionized water and water in water glass) to total solids.

**Table 3 ijerph-18-09960-t003:** Orthogonal experiment design and results.

Test No.	A (Alkali Solid Content)	B (Water Glass Modulus)	C (Liquid–Solid Ratio)	28 d Compressive Strength (MPa)
1	1 (7%)	1 (1.3)	1 (0.18)	82.11
2	1	2 (1.4)	2 (0.19)	83.68
3	1	3 (1.5)	3 (0.20)	83.48
4	2 (8%)	1	2	75.70
5	2	2	3	73.36
6	2	3	1	82.42
7	3 (9%)	1	3	74.46
8	3	2	1	75.15
9	3	3	2	82.24
Kj1	249.28	232.28	239.69	
Kj2	231.48	232.19	241.62	
Kj3	231.86	248.15	231.31	
kj1	83.09	77.43	79.90	
kj2	77.16	77.40	80.54	
kj3	77.29	82.72	77.10	
R	5.93	5.32	3.44	
Optimal level	A1	B3	C2	
Order	A > B > C			

Note: K_ji_ (j = A, B, C, D; i = 1, 2, 3): the sum of the compressive strength test values of the same level (i); k_ji_ (i = 1, 2, 3): the average value of the compressive strength test values of the same level (1/3 K_ji_); R: range.

**Table 4 ijerph-18-09960-t004:** The leaching concentrations of heavy metals in raw materials and solidified bodies with their critical limits.

Leaching Protocol	Heavy Metals	Samples							Critical Limits
		LZSS	COPR	LZC0	LZC10	LZC20	LZC30	LZC40	
TCLP	Zn (mg/L)	124.63	0.34	40.30	37.01	44.84	57.41	75.17	/
	Cr (mg/L)	0.71	54.57	0.13	0.15	0.18	0.19	0.24	5
Sulfuric acid and nitric acid method	Zn (mg/L)	18.09	0.17	0.92	0.37	1.29	1.09	1.05	100
	Cr (mg/L)	0.75	96.97	0.72	0.36	0.44	0.97	2.25	15

**Table 5 ijerph-18-09960-t005:** Elemental composition determined by EDS.

Samples	Elemental Composition (Atomic %)								
	C	O	Si	Al	Ca	Na	Fe	Zn	Cr
Spot 1	18.7	56.4	8.8	2.4	2.1	4.9	4.4	0.4	0.1
Spot 2	13.7	38.9	13.3	3.1	6	3.5	17	0.8	0.7
Area e	14.1	51.0	10.5	2.9	3.8	5.2	9.5	0.6	0.2
Area f	9.1	30.6	6.6	8.9	1.6	6.3	25.3	0.5	5.3

## Data Availability

Yu, Lin et al. (2021), The Utilization of Alkali-Activated Lead–Zinc Smelting Slag for Chromite Ore Processing Residue Solidification/Stabilization, Dryad, Dataset, https://doi.org/10.5061/dryad.zpc866t93.

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
