# Peer review of "The Utilization of Alkali-Activated Lead–Zinc Smelting Slag for Chromite Ore Processing Residue Solidification/Stabilization"

_ijerph, 2021, doi:10.3390/ijerph18199960_

Round 1
Reviewer 1 Report
Manuscript number: IJERPH-1354848-peer-review-v1
This study has shown the possibility of waste utilisation by alkali-activated lead-zinc smelting slag for chromite ore processing residue solidification/stabilization. The authors optimised the ratio of each material required to form a stable and strong solidified material, tested the leachability and evaluated the highest compressive strength of alkali-activated lead-zinc smelting slag. This work can be published after minor corrections stated below.
Keywords: start with a capital letter
Fig. 4: The authors need to clearly explain why the peaks for LZSS compared with COPR have shown less crystalline character.
L318: Note that once ZnCr2O4 formed, there are two adsorption peaks for tetrahedral and octahedral Zn-O and Cr-O. However, the octahedral was not seen since it occurs at a lower energy level. In this work, the author gave more emphasis only on the Si-O and Al-O stretching neglecting the other metal oxides and need to be included.
L319. Full stop after the citation but not before the citation.
Author Response
Responses to Reviewer #1:
Point 1: Keywords: start with a capital letter.
Response 1: We feel sorry about our mistakes. The modifications have been done accordingly. (Line: 25-26)
Point 2: Fig. 4: The authors need to clearly explain why the peaks for LZSS compared with COPR have shown less crystalline character.
Response 2: Thank you for your advice. We have added some details about the crystallographic analysis of LZSS and COPR. (Line: 303-304)
Point 3: L318: Note that once ZnCr2O4 formed, there are two adsorption peaks for tetrahedral and octahedral Zn-O and Cr-O. However, the octahedral was not seen since it occurs at a lower energy level. In this work, the author gave more emphasis only on the Si-O and Al-O stretching neglecting the other metal oxides and need to be included.
Response 3: Thank you for your valuable comments. We have deleted the incorrect reference and rewritten this part. (Line: 313-322)
Point 4: L319. Full stop after the citation but not before the citation.
Response 4: We feel sorry about our mistakes. We have reorganized the citation format of the references. (Line: 208, 344,346)
In all, your comments are quite helpful and important. We believe that your suggestion will be helpful for our future work.
Reviewer 2 Report
The article is interesting, in the investigation alkali-activated Lead-zinc smelting slag (LZSS) was used as a binder to solidify and stabilize Chromite ore processing residue (COPR). In particular it is novel and with the potential to be published, taken into account the following points:
Line 266: How many times did you perform the experiment? Why is it lower the Zn leaching concentration with LZC10 compared to LZC0? You mention that this behaviour is due to “the dissolution of raw materials in the medium and the formation of a new stable substance”. But, if it had been, this would have happened to the others too.
Line 279: At the bottom of the figure, it would be good to mention again that the concentration of LZC (10, 20, 30, 40, corresponds to the concentration of COPR).
Line 290: Why does the wustite peak (4) of LZC0, (located around 62 °) disappear with additions of COPR?.
When these points are answered, the work can be published.
Author Response
Responses to Reviewer #2:
Point 1: Line 266: How many times did you perform the experiment? Why is it lower the Zn leaching concentration with LZC10 compared to LZC0? You mention that this behavior is due to the dissolution of raw materials in the medium and the formation of a new stable substance”. But, if it had been, this would have happened to the others too..
Response 1: Thank you for your valuable comments. For each group, three parallel samples were tested, and the average was taken as the final result. The Zn leaching concentration of LZC10 was lower than that of LZC0 in both leaching methods. This may be due to the higher content of Al2O3 in COPR (29.88%) optimizing the ratio of LZSS; however, too high content leads to incomplete dissolution of aluminosilicate and hindering the condensation reaction, thus making the leaching concentration gradually increase. (Line: 281-284)
Point 2: Line 279: At the bottom of the figure, it would be good to mention again that the concentration of LZC (10, 20, 30, 40, corresponds to the concentration of COPR).
Response 2: Thank you for your advice. The modifications have been done according to your advice. (Line: 295-296)
Point 3: Line 290: Why does the wustite peak (4) of LZC0, (located around 62 °) disappear with additions of COPR?.
Response 3: We retested and analyzed the XRD patterns of LZC0-LZC40 repeatedly. We have deleted the incorrect statement and re-written this part (Line: 311-320).
In all, your comments are quite helpful and important. We believe that your suggestion will be helpful for our future work.
Reviewer 3 Report
Summary
The paper deals with the solidification/stabilization of COPR using LZSS. The subject is interesting and the paper is rather well organized overall. Some remarks are presented hereafter for the revision of the paper.
Broad comments
- In the introduction paragraph of L59, it would be interesting to specify the applications of chromium salts and chromium of which COPR is a by-product
- L64 "Since high-calcium roasting generated up to three tons of resi-63 due per ton of sodium dichromate product, it has been banned in China [20]": specify in terms of year.
- In the section methods 2.2.1. Several explanations and protocols are missing in order to clarify the section. Detailed remarks are given in the specific comments.
- Finally, optimal combinations are indicated following different tests to follow the impact of 4 parameters in the formulation but these optimal combinations have not been tested in order to verify if the sum of the formulations with 1 optimal parameter allows effectively a formulation with the 4 so-called optimal individual parameters.
- The calculation of the parameters Kij and kji is not clear in part 3.1. The interest and discussion of these parameters is not explained.
- The leaching tests have not been followed in time. What about stabilization? At what age were the samples tested? Because there can be a late stabilization or on the contrary a destabilization with time.
- The imaging part needs some improvements in the analysis in order to better understand the changes of microstructures according to the formulations
- L360: Finally, it is preferable to use a proportion of up to 20% COPR in order to maintain sufficient mechanical strength for construction applications. Beyond that, it allows at least landfilled purposes. Maybe consider re-writing this paragraph to reflect what has been well explained in the results and discussion.
Specific comments
$2.2.1.
-L102 and 103: “certain period”, need more precision.
-L106: “the prepared 105 samples were cured at 35 ℃ for 1 day”, Can you explain why?
-L107 : curing at room temperature for a certain time (6 days or 27 days)
Why these different testing times?
-L107 :” Single-factor and or-107 thogonal experiments were evaluated”: Specify the protocol or cite a reference, the Orthogonal protocol lacks explanation of the formulations studied.
$3.1.1.
-L160-163 : The paragraph should be placed after the presentation of the figure and the results, in explanation and discussion. It is placed in the wrong order.
$3.1.2.
-L223-228 : the paragraph is to be put in the method part with table 3 to explain the method. It is about the protocol and not the results.
-L231 : “the optimal combination was A1B3C2”: according to the table you have not tested this formulation so you can only say “the optimal combination could be ... according to the results on the optimal individual parameters.
-L236 : This part is really not clear as said in the broad comments. What does j mean?
$3.2.1.
- You have to specify the percentages of the COPR content in the text if you write the compressive strengths in detail with "respectively". Otherwise we don't know what percentage of COPR the strengths are related to.
- L259 : “… exceeded the criterion thresholds”: Please specify the standard take into account, thresholds depending on the country.
- L272: “…more pores…” observable at which scale? Have you some images.
- LZC notations have not been defined in the protocols.
$3.4.
- L325 mental ions
$3.5
- SEM images are too small. Add arrows on the figures to show what you explain in the text.
- L343 : « area f » : Where are these areas? In which samples are they found? It is not clear in the caption and in the text.
Author Response
Responses to Reviewer #3:
Broad comments
Point 1: In the introduction paragraph of L59, it would be interesting to specify the applications of chromium salts and chromium of which COPR is a by-product
Response 1: Thank you for your advice. We added some details in this section. (Line: 59-60)
Point 2: L64 "Since high-calcium roasting generated up to three tons of residue per ton of sodium dichromate product, it has been banned in China [20]": specify in terms of year
Response 2: Thank you for your advice. We have specified the year in this section. (Line: 102-103)
Point 3: In the section methods 2.2.1. Several explanations and protocols are missing in order to clarify the section. Detailed remarks are given in the specific comments.
Response 3: Thank you for your advice. We added some details in this section. (Line: 103, 107-110, 113-118)
Point 4: Finally, optimal combinations are indicated following different tests to follow the impact of 4 parameters in the formulation but these optimal combinations have not been tested in order to verify if the sum of the formulations with 1 optimal parameter allows effectively a formulation with the 4 so-called optimal individual parameters.
Response 4: Thank you for your advice. Single-factor experiments are the basis of orthogonal tests. Therefore, the later orthogonal experiment is further optimization of the previous single-factor experiment. (Line: 113-115)
Point 5: The calculation of the parameters Kij and kji is not clear in part 3.1. The interest and discussion of these parameters is not explained.
Response 5: Thank you for pointing out our mistakes. The modifications have been done according to your advice. (Line: 239-240)
Point 6: The leaching tests have not been followed in time. What about stabilization? At what age were the samples tested? Because there can be a late stabilization or on the contrary a destabilization with time.
Response 6: Thank you for your valuable comments. All leaching tests were performed immediately after 28 days of curing. (Line: 136-137)
Point 7: The imaging part needs some improvements in the analysis in order to better understand the changes of microstructures according to the formulations.
Response 7: Thank you for your valuable comments. The modifications have been done according to your advice. (Line: 355)
Point 8: L360: Finally, it is preferable to use a proportion of up to 20% COPR in order to maintain sufficient mechanical strength for construction applications. Beyond that, it allows at least landfilled purposes. Maybe consider re-writing this paragraph to reflect what has been well explained in the results and discussion.
Response 8: Thank you for your advice. We have re-written this paragraph. (Line: 394-403)
Specific comments
Point 9: L102 and 103: "certain period", need more precision.
Response 9: Thank you for your advice. We have specified the time in this section. (Line: 103)
Point 10: L106: "the prepared samples were cured at 35 ℃ for 1 day", Can you explain why?
Response 10: Thank you for pointing out our mistakes. We have deleted the incorrect statement and re-written this part. (Line: 106-110)
Point 11: L107: curing at room temperature for a certain time (6 days or 27 days). Why these different testing times?
Response 11: Thank you for your comment. Combined with the initial curing time of 1 day, the test is for compressive strength at 7 and 28 days. The following reference also took this method:
https://doi.org/10.1016/j.jclepro.2018.10.265
Point 12: L107: Single-factor and orthogonal experiments were evaluated": Specify the protocol or cite a reference, the Orthogonal protocol lacks explanation of the formulations studied.
Response 12: Thank you for your advice. We have re-written this part according to your comment. (Line: 113-118)
Point 13: $3.1.1. L160-163: The paragraph should be placed after the presentation of the figure and the results, in explanation and discussion. It is placed in the wrong order.
Response 13: Thank you for your advice. The modifications have been done according to your advice. (Line: 250-251)
Point 14: $3.1.2. L223-228: the paragraph is to be put in the method part with table 3 to explain the method. It is about the protocol and not the results.
Response 14: Thank you for your advice. The modifications have been done according to your advice. (Line: 113-118)
Point 15: $3.1.2. L231: "the optimal combination was A1B3C2": according to the table you have not tested this formulation so you can only say "the optimal combination could be ... according to the results on the optimal individual parameters.
Response 15: Thank you for your advice. The modifications have been done according to your advice. (Line: 244-247)
Point 16: $3.1.2. L236: This part is really not clear as said in the broad comments. What does j mean?
Response 16: Thank you for your comment. "j" represents the different factors (A, B, C) in the orthogonal experiment. (Line: 239-240)
Point 17: $3.2.1. You have to specify the percentages of the COPR content in the text if you write the compressive strengths in detail with "respectively". Otherwise, we don't know what percentage of COPR the strengths are related to.
Response 17: Thank you for your advice. The modifications have been done according to your advice. (Line: 254-256, 259-261)
Point 18: $3.2.1. "… exceeded the criterion thresholds": Please specify the standard take into account, thresholds depending on the country.
Response 18: Thank you for your comment. We have specified the standard in this section. (Line: 270-272)
Point 19: $3.2.1. "…more pores…" observable at which scale? Have you some images.
Response 19: We feel sorry about our mistakes. In addition, we have deleted the incorrect statement and re-written this part.
Point 20: $3.2.1. LZC notations have not been defined in the protocols.
Response 20: We feel sorry about our mistakes. We had defined LZC when it first appeared in the article. (Line: 85)
Point 21: $3.4. mental ions.
Response 21: We feel sorry about our mistakes. We changed the "mental ions" into "metal ions" in the manuscript. (Line: 352)
Point 22: $3.5. SEM images are too small. Add arrows on the figures to show what you explain in the text.
Response 22: Thank you for your advice. We have adjusted the SEM images. (Line: 355)
Point 23: $3.5. L343: «area f»: Where are these areas? In which samples are they found? It is not clear in the caption and in the text.
Response 23: We feel sorry about our fault. We have adjusted the SEM images. (Line: 355)
In all, your comments are quite helpful and important. We believe that your suggestion will be helpful for our future work.
Round 2
Reviewer 3 Report
The amendments by the authors have clarified the information in this manuscript.
It would be relevant prior to publication to include the references in the bibliography corresponding to responses 1, 11 and 18.
Author Response
Dear reviewer,
We would like to express our sincere appreciation for your careful reading and helpful comments. We have addressed the points. In addition, the revised portion were marked using red color, and the deletions were marked with green strikethrough in "Revised Manuscript " document.
Point 1: It would be relevant prior to publication to include the references in the bibliography corresponding to responses 1, 11 and 18.
Response 1: Thank you for your advice. We have added references as you suggested. (Line: 59, 107, 260-261)
Thanks again for your valuable time and comments.
Yours sincerely,
Prof. & Dr. Dongwei Li